# Effects of Degassing Treatment on the Dielectric Properties of XLPE Insulation Used in High-Voltage DC Power Cables

**DOI:** 10.3390/polym17030431

**Published:** 2025-02-06

**Authors:** Man Ding, Qingfeng Zheng, Jiahe Wang, Weifeng He, Chao Dai, Dingjun Wen

**Affiliations:** 1School of Electrical and Power Engineering, Hohai University, Nanjing 211100, China; ding.m@hhu.edu.cn (M.D.); qfzheng0922@foxmail.com (Q.Z.);; 2Research Institute, State Grid Gansu Electric Power Co., Ltd., Lanzhou 730071, China; 3State Grid Yangzhou Electric Power Supply Company, Yangzhou 225000, China; wangjiahe@hhu.edu.cn

**Keywords:** XLPE, power cable, cross-linking by-product, degassing treatment

## Abstract

Cross-linked polyethylene power cables are widely used in high-voltage DC transmission lines, owing to their good dielectric and physical–chemical properties. However, the production process of XLPE involves cross-linking and degassing, in which the cross-linking process produces a variety of cross-linking by-products, and the changes in the properties of the cable insulation caused by the degassing process are not well understood. XLPE samples were degassed at 90 °C for 7 and 14 days in this paper, and the main by-products were found to be α-methylstyrene, acetophenone, and cumyl alcohol, the contents of which all declined after the degassing treatment. The results show that the space charge density, the leakage current under a high electric field at different temperatures, and the breakdown strength of the XLPE samples all decreased after the degassing treatment. On the other hand, the XLPE sample after 7 days’ degassing had the lowest conductivity and the highest conductance activation, and the space charge density and the charge decay rate as well as the breakdown strength after 7 days’ degassing differed little from the 14-day treated sample, demonstrating that the 7-day degassing treatment at 90 °C would be enough to achieve superior performance.

## 1. Introduction

High-voltage DC transmission is widely used in new energy power scale applications and long-distance large-capacity transmission projects, such as cross-regional DC power grids, offshore wind power access, power transmission in alpine areas, and island power supply [1]. It is more flexible and reliable, and is more low-carbon and environmentally friendly, meaning it helps to improve the security and reliability of the transmission network capacity increase, and enhances the ability to withstand disasters in the process of urbanization [2,3,4,5,6].

Cross-linked polyethylene (XLPE) has good dielectric properties and resistivity, as well as stable physical–chemical properties, including high resistance to cracking and moisture, economical production, and being environmentally friendly, and it is currently the most commonly used high-voltage cable insulation material in the power industry worldwide. Crosslinking technology is used to change polyethylene from a linear chain polymer thermoplastic material to a thermosetting material with a cross-linked molecular chain three-dimensional network structure [7,8,9]. Currently, the main cross-linking methods for XLPE are silane cross-linking, used in medium- and low-voltage power cables and electrical equipment cables; irradiation cross-linking, used in low-voltage cable insulation and jacketing; and peroxide chemical cross-linking, used in high-voltage and ultra-high-voltage power cables. Disopropyl peroxide (DCP) is a commonly used cross-linking agent, which produces a variety of cross-linking by-products including acetophenone, chlortetrol, and α-methylstyrene during thermal decomposition. The polar molecules in the by-products may lead to space charge accumulation and electrical field distortion, and even the aging or partial discharge and breakdown of XLPE in severe cases. As such, a degassing treatment is necessary after the cross-linking process in order to eliminate the impacts of the by-products on the dielectric properties of XLPE [10,11,12].

The design of high-voltage DC cables mainly refers to the design principles of AC cables, and relatively conservative insulation thickness and degassing process design principles are usually chosen in order to ensure the reliable operation of the cables. The cable degassing process is still mainly experimental, as there are no clear regulations and standards for degassing parameters to ensure maximized industrial efficiency. For 500 kV HVDC cables, a degassing time of up to dozens of days causes a substantial increase in production costs, prolonging the commissioning of the cable costs and prolonging the commissioning time of the cables [13]. The insulation properties, microstructural changes, and related mechanisms of the materials in the degassing process are not well understood. Moreover, we lack research on the insulation properties and micromorphological characteristics of 500 kV XLPE for HVDC during the degassing process, and the quantitative relationships between the DC breakdown, the conductivity, the space charge characteristics of the material, and the dissociation of the crosslinking by-products are not clear. So, it is of great theoretical significance and engineering value to carry out a study on the effects of degassing on the performance of domestically produced XLPE HVDC cable materials.

Recently, the influence of degassing treatment on the insulating property of cable insulation has been studied by many researchers, especially regarding the influence on the cross-linking by-product content, space charge accumulation, and the dielectric properties. From the results regarding the change in cross-linking by-product content with degassing treatment time, degassing was thought to be helpful in dielectric and mechanical property improvement [11,12,13,14,15,16]. However, the microscopic change mechanisms of the impact of degassing are not definite. The cross-linking by-products decompose into polar micromolecules during cable operation and then dissociate into positive–negative ion pairs under an electric field. These ion pairs transport to operate electrodes and form hetero space charges on the edge of the cable insulation, leading to electrical field distortion [13,16,17,18,19,20,21,22,23,24,25]. Degassing treatment was found to be able to improve the electrical strength of XLPE and weaken the dependence of the breakdown field on the XLPE insulation thickness [11]. Moreover, some researchers have studied the measurement methods of the degassing process of cross-linking by-products of XLPE, and weight loss and HPLC (high-pressure liquid chromatography) have shown themselves to be the most effective and practical measurement techniques [7]. 

The influence of degassing on the microscopic structure of XLPE has also been studied. In the initial stage of degassing, elongation at break increases with the change of the chemical chains of XLPE owing to re-crosslinking. Meanwhile, crystal nucleus formation and grain growth take place under the thermal effect of degassing and the nucleation impacts of by-products, which also leads to an increase in elongation at break, and the crystallinity increases owing to the perfection of sphere crystals. As degassing proceeds, the re-crosslinking, grain growth, and crystal nucleus formation are inhibited as the cross-linker runs out and the by-products volatize, after which the elongation at break, sphere crystal size, and crystallinity then tend to stabilize [21].

The influence of degassing treatment on the aggregation structure, electrical tree characteristic, charge transportation, and the space charge distribution and evolution properties of XLPE cable insulation has been studied by some researchers. For XLPE used as AC cable insulation, degassing treatment was found to inhibit the growth of electrical trees caused by voids and the cross-linking by-product acting as a voltage stabilizer [26]. The influence of degassing treatment on the dielectric properties of ultra-pure XLPE material was investigated in [13,20], which revealed that degassing treatment facilitates the volatility of by-products. Samples under 36 h of degassing exhibited the highest breakdown strength and lowest DC conductivity, and improvements were also noted in space charge accumulation and electrical field distortion characteristics after the degassing treatment. Additionally, the XLPE insulation of the operational 525 kV DC cable was studied to determine the effect of degassing in [17]. It was found that the contents of cross-linking by-products decreased with increased treatment time, and stabilized after 30 days. The dissociation of these by-products leads hetero-charge accumulation within XLPE insulation. However, the effect of degassing treatment on domestic XLPE materials has not yet been studied, and changes in aggregation structure, charge transport, and space charge accumulation after degassing have not been clarified.

The degassing process after the production of cross-linked polyethylene cables is often not thorough enough, resulting in a certain amount of cross-linking by-products remaining in the main insulation when the cable is put into operation. The heat generated by the conductor during cable operation under an electric field promotes the volatilization of these cross-linking by-products, effectively continuing the degassing process during the early stage of cable operation. In this study, the change in the content of cross-linked by-products within the main insulation of XLPE and their effects on the space charge characteristics, DC conductivity and DC electrical strength are studied by simulating the degassing process caused by thermal stress during cable operation. The XLPE material used in this study was provided by a domestic cable material factory. The influences of the degassing treatment on the DC conductivity and breakdown strength of XLPE were studied, and the effects of the dissociation of cross-linking by-products on space charge and their dynamic behavior were also clarified; the optimal degassing duration was detected. The findings from this study are beneficial for the domestic development of ultra-pure cable materials and the reliable operation of high-voltage DC cables.

## 2. Sample Preparation and Characterization

### 2.1. Sample Preparation and Degassing Treatment

The granular XLPE raw material used in this study was purchased from Wanma Cable Co. in Hangzhou, Zhejiang, China, and is intended for used in 330 kV HVDC cable. Circular film samples with a diameter of 100 mm and varying thicknesses were prepared using a hot-pressing method with a plate vulcanization machine at 15 MPa and 180 °C. Prior to testing, the film samples were dried for 24 h in a vacuum oven at 80 °C to eliminate the effects of mechanical stress induced during the hot pressing. Since the long-term operating temperature of the XLPE cable is approximately 90 °C, 90 °C was selected as the degassing treatment temperature in this study. The samples were divided into three groups and then degassed at 90 °C for different durations to investigate the effects of degassing treatment on the dielectric properties. In accordance with the conditions recommended by GB/T 11017.1-2014 [27] and GB/T 2951.12-2008 [28], the degassing hdurations were set to 0 days, 7 days and 14 days, respectively. After degassing, the insulation properties were examined.

### 2.2. Sample Characterization

Thermo Scientific Nicolet iS5 Fourier transform infrared spectroscopy (FTIR) from Waltham, MA, USA was used to test the chemical structures of XLPE films with wave numbers ranging from 500 cm^−1^ to 4000 cm^−1^ under attenuated total reflectance (ATR) mode. From the obtained spectra, the molecular chain characteristics and typical functional groups in XLPE samples were identified.

The leakage current of the XLPE film was measured using Keithley 6517B (from Everett, Washington, DC, USA) based on a three-electrode system. Stepped DC voltage was applied to the samples that were placed in a temperature-controlled chamber. The testing temperature was maintained at 25 °C with an accuracy of ±1 °C. The measurement duration for each sample was 30 min.

The space charge characteristic of XLPE films was measured using the pulsed electro-acoustic (PEA) method. A semiconductor (carbon-loaded polyethylene) and aluminum were used as the upper and bottom electrodes, respectively, with the XLPE film sandwiched between them, and silicon oil was used as the acoustic couplant. Space charge profiles were measured at an electric field of 50 kV/mm under room temperature. Since previous studies have shown that the space charge characteristics change little between 20 min and 2 h, the voltage application and test duration were set to 20 min.

A DC breakdown test was conducted to determine the electrical strength, during which a DC voltage was applied to the 0.2 mm thick XLPE film with the voltage increasing at a rate of 1–2 kV/s until breakdown occurred. The films were sandwiched between sphere-to-sphere electrodes and immersed in oil during testing to prevent surface flashover. Ten to fifteen measurements were taken for each sample, and a two-parameter Weibull statistical analysis was performed on the breakdown field data.

## 3. Experimental Results and Discussion

### 3.1. The Cross-Link By-Product Content in XLPE After Degassing Treatment

The Fourier transform infrared spectra of the XLPE samples are shown in Figure 1. The peaks located at 718 cm^−1^ and 1469 cm^−1^ correspond to –CH_2_ bending vibration absorption peaks, while the peaks at 2846 cm^−1^ and 2915 cm^−1^ correspond to the –CH_2_ stretching vibration peaks [22].

Generally, the linear molecular structure of polyethylene is chemically or physically modified into a three-dimensional network structure of XLPE to improve the thermal and mechanical properties of the material. Nowadays, the most commonly used cross-linking method is the peroxide method, which uses dicumyl peroxide (DCP) as the cross-linking agent. However, by-products are generated during the cross-linking process and remain inside the XLPE as impurities. These impurities can dissociate, generating positive and negative ions under high voltage, which migrate inside the XLPE toward opposite electrodes and form space charges near the electrodes. The space charge inside the XLPE can lead to electric field distortion and, in severe cases, even the partial discharge or breakdown of the material. Therefore, it is important to investigate the cross-linking by-products during the degassing treatment of XLPE. The absorption peaks are usually located in the ranges of 1600–1800 cm^−1^ and 3100–3600 cm^−1^, so the detailed FTIR spectra in these two typical ranges were obtained and are shown in Figure 2. The absorption peaks at 1640 cm^−1^, 1720 cm^−1^ and 3370 cm^−1^ correspond to the cross-linking by-products, including styrene, acetophenone and cumyl alchole chemical groups, respectively.

Although high-voltage power cables are degassed before assembly, the degassing time is often insufficient for all the cross-linking by-products to be released completely. This is demonstrated in Figure 2 by the absorption peaks at 1720 cm^−1^ and 3370 cm^−1^, which correspond to acetophenone and cumyl alchole chemical groups, respectively. The absorption peak at 1640 cm^−1^ is considered to indicate the styrene C = C double bond stretching vibration, which is introduced by the applied DC voltage and reflects the ageing degree of the XLPE material. To quantify the by-product contents, the absorption peak at 718 cm^−1^ was used as the reference peak. The ratios between each cross-linking by-product and the reference peak were calculated and are presented as the by-product index, as shown in Figure 3. Equation (1) is as follows:(1)$$C=\frac{I_{x}}{I_{718}}$$
where *C* is by-product index, *I_x_* is by-product absorbance peak intensity, and *I*_718_ is the absorbance peak intensity at the absorption peak at 718 cm^−1^.

All the cross-linking by-products were found to decrease with increasing treatment times, as shown in Figure 3. The content of styrene decreased by 16.1% after 7 days of degassing and by 17.6% after 14 days of degassing compared to the untreated samples. Similarly, the corresponding decreases in content were 3.0% and 20.4% for acetophenone, and 26.2% and 40.0% for cumyl alcohol. Since the longest degassing time in this study was 14 days, the change in by-product content due to the thermal effect during cable operation was not taken into consideration [13].

The cross-linking by-products are present inside XLPE as small molecules that act as impurities. These molecules can dissociate and migrate within the XLPE under the electric field, affecting the conductivity of the material. Additionally, these small molecules can act as traps centers, leading to charge trapping and the formation of space charges inside the XLPE. The reduction in these by-products could alter either the DC conductivity or the space charge accumulation in XLPE, as will be demonstrated in the following sections.

### 3.2. The Space Charge Characteristic of XLPE Films with Different Degassing Times

The XLPE films were placed in a vacuum oven at 80 °C for 48 h to ensure uniform initial thermal and mechanical conditions. Space charge measurements were conducted using a PEA measurement system. Both the upper and bottom electrodes are made of aluminum, with a semiconductor (carbon black loaded polyethylene) layer attached to the upper electrode. The 0.2 mm thick XLPE films were measured under an electric field of 50 kV/mm at room temperature. Each measurement lasted for 20 min until reaching a steady or quasi-steady state. Previous studies have shown that the space charge distribution changes little after 20 min compared to that after 2 h, suggesting that most of the space charge dynamics occur within the first 20 min. Considering this, all space charge tests in this paper were performed for 20 min, followed by a 10-min short-circuit period. The space charge distribution characteristics of the three groups of XLPE samples are presented in Figure 4.

It can be seen from Figure 4 that there is an obvious charge accumulation, and the carrier injection from the anode and cathode increases with time. This leads to the formation of space charge packets. Specifically, the space charge accumulation shifts from near the electrodes to the interior of the XLPE film, indicating that the space charge packet formation migrates towards the inner material as time progresses under high voltage. This migration distorts the local electric field distribution and makes it easier to break down. Specifically, positive charges and negative charges accumulate on the anode and cathode under 50 kV/mm, after 20 min, respectively.

To investigate the space charge distribution and its changes inside XLPE, the charge dissipation process was tested, and is shown in Figure 5.

Figure 5 shows the space charge distribution of XLPE samples after different degassing treatment times during the depolarization test. For the untreated sample, homo-charges accumulated near the electrodes, with positive charges near the positive electrode and negative charges near the negative electrode after 10 s of depolarization. The amount of space charge near the electrodes decayed significantly, while the charge amount inside the sample changed little with increasing depolarization time. The space charge distributions at 300 s and 600 s differed little, indicating that charges reached equilibrium at approximately 300 s. Additionally, the charge quantity near the electrodes decreased little after 7 days of degassing compared to the untreated sample. However, after 14 days of degassing, the charge quantity near the electrodes increased, while the charge quantity inside the sample decreased significantly. This suggests that most space charges concentrate at the interface between the electrode and the XLPE film.

From the results in Section 3.1, we see that charges accumulated near the electrodes due to the ion migration caused by the dissociation of cross-linking by-products under a large electric field. Therefore, the charge density would decrease with the reduction in these by-products after degassing, implying that degassing treatment suppresses the charge accumulation near the electrodes.

To study the influence of degassing on the space charge accumulation characteristic inside XLPE, the dynamic response of space charge was analyzed and calculated using Equation (2) [24,25].(2)$$q(t,E_{p})=\frac{1}{x_{1}-x_{0}}\int_{x_{0}}^{x_{1}}\left|q_{p}(x,t,E_{p})\right|dx$$
where *x* is the location inside the sample, *x*_1_ and *x*_0_ represent the locations of upper and bottom electrodes, *q_p_*(*x*, *t*, *E_p_*) is the space charge density inside XLPE, *t* is the polarization time, and *E_p_* is the polarization electric field. In order to quantitatively analyze the total charge amount, the absolute value of space charge was used during calculation, and the results are shown in Figure 6.

From Figure 6a, we see that the space charge amount accumulating inside the sample increased gradually over time during polarization, and the charge amount reduced significantly with the degassing time. From Figure 6b, we see that the space charge accumulation declined dramatically at the beginning of depolarization and reached an equilibrium state after a very short time. The charge density during depolarization decreased with the degassing time, with the value of 2.3242 C/m^3^ for the untreated sample, 0.8888 C/m^3^ for the 7-days degassing sample and 0.8736 C/m^3^ for the 14-days degassing sample.

The decay rate of the space charge inside XLPE was analyzed based on the space charge measurement results using Equation (3).(3)$$v=\frac{\Delta q(t,E_{p})}{\Delta t}=\frac{\left|q(t_{2},E_{p})-q(t_{1},E_{p})\right|}{t_{2}-t_{1}}$$
where *t*_1_ and *t*_2_ are depolarization times, and Δ*t* is the difference between them; *q(t, E_p_)* is the space charge density at time *t* under the electric field of *Ep*, and Δ*q(t, E_p_)* is the difference of charge density in the time interval.

The space charge decay rate under the electric field of 50 kV/mm in the time interval of 0 s to 600 s was calculated, and the results are shown in Table 1.

The cross-linking by-product content decreased after the degassing treatment from the result shown in Section 3.1. This reduction implies that the quantity of positive–negative ion pairs and the amount of charge trapped by shallow traps, which result from these by-products and their dissociation, also decreased correspondingly. Consequently, there were fewer heteropolar ion recombinations and less charge detrapping during the short-circuit depolarization process when the by-product content was reduced. The space charge decay rate was lower when there were fewer ion pair recombinations and less charge detrapping. Therefore, the XLPE sample after 14 days of degassing exhibited the lowest decay rate, as shown in Table 2, because it had the lowest by-product content, as observed in Figure 3.

From Section 3.1, it is evident that the cross-linking by-product content in XLPE can be reduced through degassing treatment, which subsequently suppresses space charge accumulation. These by-products act as impurities and create a large number of charge traps, affecting the charge transport process by trapping charge carriers. To assess the influence of these by-products on the charge transport characteristics in XLPE, the DC conductivity of XLPE samples before and after degassing treatment was tested.

### 3.3. DC Conductivity Characteristic

The DC leakage current characteristics of XLPE films with different degassing treatment times are shown in Figure 7.

The leakage current under different temperatures changes little after either 7 days or 14 days of degassing treatment when the electric field is relatively low. However, it decreases obviously after degassing under a higher filed. Specifically, the difference in leakage current between films degassed for 7 days and those degassed for 14 days’ is comparatively smaller than that observed between degassed films and untreated films. Additionally, the behavior of the leakage current with an increasing electric field can be divided into two distinctive regions. In area I, the leakage current shows minimal change with increasing electric field. In contrast, in area II, the leakage current increases sharply as the electric field increases.

The straight lines in Figure 7 are fitted curves of the leakage currents, which exhibit different slopes and field thresholds when transitioning from area I to area II. The slopes and the corresponding field thresholds at each temperature are detailed in Table 2.

From Table 2, it is evident that the slopes of the fitting curves in area I increase after degassing treatment, suggesting that the XLPE samples become more sensitive to the electric field following degassing. Furthermore, the electric field thresholds for transitioning from area I to area II increase with longer degassing time under each temperature. This indicates that the field sensitivity of the XLPE improves as a result of degassing.

The DC conductivity of XLPE has a great influence on the operational performance of DC power cables, which can be calculated from the leakage current by using Equation (4).(4)$$\sigma =\frac{J}{E}=\frac{Id}{SU}$$
where *σ* (S/m) is the DC conductivity, *J* (A/m^2^) is the conductive current per unit area, *E* (V/m) is the applied electric field strength during testing, *I* (A) is the conductive current, *S* (m^2^) is the testing area, *d* (m) is the tested sample thickness, *U* (V) is the applied voltage during testing. The DC conductivities of each of the XLPE samples were figured out and are shown in Figure 8.

From Figure 8, the DC conductivities of XLPE samples before and after degassing are in the order of 10^−16^–10^−15^ S/m magnitude, which is in accordance with the empirical conductivity values of XLPE. The DC conductivity increases with rising temperature. Moreover, we can see that the conductivity of XLPE decreases after 7 days’ degassing treatment, and then slightly increases after 14 days’ degassing at each temperature. Compared to the XLPE sample before degassing, the DC conductivity of the samples after 7 days of degassing decreases by 48.5%, 44.3% and 15.4% at 30 °C, 50 °C and 70 °C, respectively; for samples after 14 days of degassing, the decreases in conductivity are 29.3%, 35.3% and 5.84% at the same temperatures.

Moreover, the DC conductivity determines the steady-state electric field distribution within the XLPE insulation of DC cables. The temperature dependence of the conductivity significantly affects the radial electric field distribution in HVDC power cables. This temperature dependence is characterized by the activation energy, as shown in Equation (5).(5)$$\sigma =A\exp(-\frac{E_{i}}{kT})\frac{\sinh(B|E|)}{\left|E\right|}$$
where *σ* (S/m) is the DC conductivity, *A* is a constant determined only by the material, *k* (eV/K) is the Boltzmann constant, *T* (K) is the kelvin temperature, *E_i_* (eV) is the conductance activation energy, B is the dependence coefficient of the conductivity on the electric field, and *E* (V/m) is the applied electric field strength.

The logarithm of conductivity is found to be linearly dependent on the reciprocal of the Kelvin temperature, as described by the Arrhenius equation shown in Equation (4). The plot for 25 kV/mm is shown in Figure 9, and the conductance activation energy can be determined by calculating the slope of the figure. The symbols in Figure 9 are the experimental results, and the straight lines are fitting results.

The conductance activation energies of each sample were determined to be 0.74 eV, 0.85 eV, and 0.81 eV for the untreated, 7 days’ and 14 days’ degassing treatments, respectively. The conductance activation energy reflects the barrier height that carriers must overcome during the conductance process in XLPE. The higher the activation energy, the more difficult it is for carriers to overcome the barrier and participate in the conductance current. From the conductivity and conductance activation energy results, it can be concluded that the XLPE samples have the lowest conductivity and the highest activation energy after a 7-day degassing treatment compared to the other two groups of XLPE samples. Moreover, the larger conductance activation energy may be related to a greater number of deep traps in the corresponding samples, making it harder for carriers to transport through the material and reducing the leakage current.

### 3.4. DC Breakdown Characteristic

The breakdown strength of XLPE determines its ability to maintain the dielectric performance under electric stress, which reflects the insulation condition of power cables. In this study, a two-parameter Weibull distribution analysis was performed on the DC breakdown voltages of XLPE samples. The breakdown probability of the XLPE samples under high voltage can be formulized using Equation (6), according to the Weibull distribution function.(6)$$F(U,\alpha ,\beta )=1-\exp[-{\left(\frac{U}{\alpha }\right)}^{\beta }]$$
where *F* is the breakdown probability of the tested XLPE samples under voltage *U*, and *α* and *β* are scale and shape parameters of the Weibull distribution.

Figure 10 shows the breakdown characteristic following Weibull distribution, and the breakdown voltages and Weibull shape parameter are also shown in this figure.

The breakdown voltages of XLPE films were found to increase with the aging duration time. The breakdown strength of the untreated XLPE was 162.23 kV/mm, and its Weibull shape parameter was 8.98. After 7 days of degassing treatment, the breakdown strength increased to 191.50 kV/mm, which is 18% higher than that of the untreated sample. However, the Weibull shape parameter was distinctly larger than that of the untreated film. After 14 days of degassing treatment, the breakdown strength increased to 194.45 kV/mm, while the shape parameter decreased to 5.43. This implies that longer degassing times can reduce the breakdown dispersibility and further improve the insulation stability of the XLPE material. The analysis of variance (ANOVA) method was also used in the study to estimate the breakdown strength of different XLPE samples. The average breakdown fields were 154.05 kV/mm, 187.62 kV/mm and 179.67 kV/mm for the XLPE samples with degassing times of 0, 7 and 14 days, respectively. The standard deviations were 18.4, 8.7 and 37.6 for the corresponding samples. The breakdown strengths calculated from the ANOVA method differed little from those obtained using the Weibull analysis method, which aligns with the breakdown mechanism of XLPE.

## 4. Discussion

The peroxide crosslinking method used in cable manufacturing led to the generation of the by-product, as illustrated in Figure 11. The most commonly used cross-linker is dicumyl peroxide (DCP), which decomposes at high temperatures. There are two chemical reaction routes, both of which generate active free radicals that abstract hydrogen atoms from the polyethylene macromolecular chains, forming cross-linked bonds and promoting the reticulation of the structure. Two primary by-products, acetophenone and cumyl alcohol, are generated through these two reaction routes, along with other by-products such as α-methyl styrene, methane and H_2_O, etc. [12]. Methane is gaseous at room temperature and easily escapes from XLPE. The remaining by-products, such as acetophenone, cumyl alcohol and α-methyl styrene, are polar molecules that can dissociate under thermal stress. This dissociation barrier can be lowered under an applied electrical field. The charges released from this dissociation can strengthen the local electric field and reduce the insulating properties, affecting the insulation life of XLPE and threatening the operation of power cables. To minimize the amount of cross-linking by-products in the cable, XLPE cables are generally degassed by placing them in a high-temperature vacuum atmosphere [13].

In this study, XLPE samples were degassed at 90 °C for 7 days and 14 days. The main by-products identified were α-methylstyrene, acetophenone, carbonyl and cumyl alcohol, and the contents of these by-products all declined to various degrees after degassing treatment. Fewer by-products implies lower space charge accumulation near the electrodes and reduced space charge amounts during polarization and depolarization, as verified by the space charge characteristic results shown in Figure 4, Figure 5 and Figure 6. Moreover, fewer by-products result in less charge transportation inside the XLPE, which is consistent with the DC conductive characteristics shown in Figure 7 and Figure 8. Styrene, cumyl alcohol and acetophenone exist in the form of micro-molecules that enhance the conductivity in XLPE. The degassing treatment facilitates the volatilization of these micro-molecule by-products, as shown in Figure 3. Consequently, the leakage current of XLPE decreases with the reduction in by-product contents, a finding that aligns with the results reported in [13]. Furthermore, the conductance activation energy was lowered after degassing, which can be attributed to the reduction in shallow traps cause by the by-products, thereby assisting charge transport [26].

The relationship between the leakage current and applied electric field in XLPE can be divided into two areas as the electric field increases. In the low field area, the current in XLPE with different degassing treatments changes little after degassing, while it increases obviously after degassing under each electric field and temperature. The leakage current was reported to follow Ohm’s law and the space charge limited current (SCLC) law in [15]. There is a threshold field between the two current areas. When the electric field is low, leakage current depends primarily on the applied field rather than the traps inside XLPE, and thus the current dose not rise substantially with the electric field, resulting in a small slope in the relationship curve shown in Figure 7. When the electric field exceeds the threshold field, the charge transport characteristics in XLPE are determined by both the electric field and the traps within the material. If the amount of charge injected from the electrode into XLPE exceeds the charge carrier transport capacity inside XLPE, space charges accumulate near the electrode, reducing the electric field and restraining further charge injection from the electrodes. The leakage current reaches a dynamic equilibrium state when the charge injection rate from the electrode to XLPE equals the charge transport rate inside XLPE. This current is referred to as space charge limited current. In this high field area, traps within XLPE play a significant role in the charge transport process, and their depth and distribution affect the current. The decline in leakage current observed in the high field area in Figure 7 can be attributed to changes in trap distribution following degassing treatment, which alters the by-product content. Moreover, although the XLPE sample after 7 days of degassing has the lowest conductivity and the highest conductance activation energy, these values differ only slightly from those of XLPE after 14 days of degassing. During degassing treatment, a large number of by-products dissociate, introducing more traps into the dielectric. Even though the carrier migration rate increases with temperature and electric field, many carriers become trapped during migration and cannot participate in the conductance process. This explains why the conductivity of XLPE after 14 days of degassing is slightly larger than that after 7 days of degassing.

Regarding the space charge distribution, positive space charges accumulate near the cathode in samples after 7 days of degassing, while negative charges accumulate near the cathode in samples after 14 days of degassing, as shown in Figure 4. From the perspective of the total charge amount inside bulk XLPE, the charge amount decreases sharply after 7 days of degassing and then changes little after 14 days of treatment. The charge decay rate also declines with increasing degassing treatment time, as shown in Figure 6 and Figure 7, which is consistent with the by-product content shown in Figure 3. The dissipation of by-products leads to a decrease in trap density, and the energy level shifts to lower values, suppressing the accumulation of space charges. This result is in accordance with findings reported in [20].

On the other hand, the aggregated structure change analysis of XLPE after degassing treatment in [16] shows that degassing treatment can promote the crystallization and improve the crystallinity of XLPE. The molecule chains in the amorphous regions fold in an orderly way to achieve recrystallization under thermal stress, leading to an increase in lamellar thickness and an improvement in the crystal structure. During the orderly folding and movement of molecule chains, voids are generated, forming new traps that can capture charges, making charge trapping more likely. As a result, the carrier migration rate decreases, contributing to an increase in breakdown strength due to the improved crystal structure.

In conclusion, the degassing treatment of XLPE can promote the dissipation of cross-linking by-products, thereby reducing space charge density, since by-product dissociation is the origin of space charge accumulation. On the other hand, degassing treatment can enhance the aggregation structure of XLPE and promote the reordering of molecule chains, which increases trap density. These traps capture charge carriers and suppress charge transport within XLPE, leading to a reduction in the leakage current and conductivity. However, when the degassing time increases to 14 days, the aggregation structure may be broken, creating more charge transport channels and slightly increasing the leakage current. The breakdown strength of XLPE increases after degassing due to the improved aggregation structure and reduced electrical field distortion caused by less space charge accumulation. Therefore, for the domestic XLPE cable insulation material, a degassing treatment duration of 7 days at 90 °C would be sufficient.

## 5. Conclusions

The influence of the degassing treatment on the cross-linking by-products and their effects on charge transportation and charge accumulation characteristics in XLPE cable insulation were studied in this paper. The main conclusions are as follows:

(1) Cross-linking by-products, including styrene, acetophenone and cumyl alchole, were identified in XLPE samples. Their contents were found to decrease with increasing degassing treatment time. For XLPE samples treated for 14 days, the largest decrease was observed in cumyl alcohol, with a reduction rate as high as 40%.

(2) The charge density during depolarization, the leakage current under high electric fields at different temperatures, and the breakdown strength of the XLPE samples all decrease to varying degrees after 7 and 14 days of degassing treatment. This demonstrates that space charge accumulation inside XLPE can be inhibited by the degassing treatment due to reduced hetero polarity ion recombination and charge de-trapping. Moreover, the decrease in leakage current and improvement in breakdown strength can be attributed to changes in trap distribution and an enhanced aggregated structure.

(3) The XLPE sample after 7 days of degassing exhibited the lowest conductivity and the highest conductance activation energy, although these values differed only slightly from those of the XLPE sample after 14 days of degassing. The charge amount was found to decrease sharply after 7 days of degassing and then changed little after 14 days of treatment. Similarly, the charge decay rate followed the same trend with increasing degassing treatment time, implying that a 7-day degassing treatment at 90 °C would be sufficient to achieve superior performance.

(4) Degassing treatment can improve the charge transport, space charge accumulation characteristic, and breakdown strength by dissipating cross-linking by-products. A 7-day degassing treatment at 90 °C is sufficient for a domestic XLPE material. These results are beneficial for the localization of ultra-pure XLPE materials and their application in high-voltage DC power cables.

## Figures and Tables

**Figure 1 polymers-17-00431-f001:**
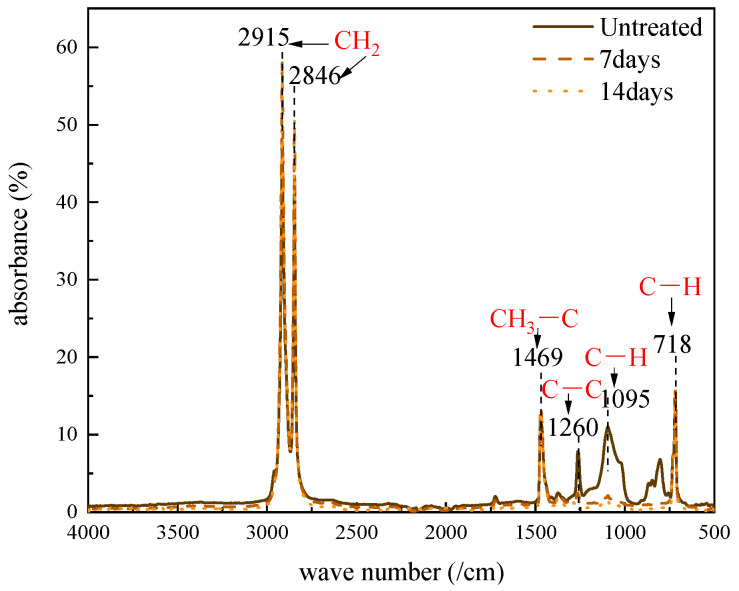
Fourier transform infrared spectrum of XLPE film.

**Figure 2 polymers-17-00431-f002:**
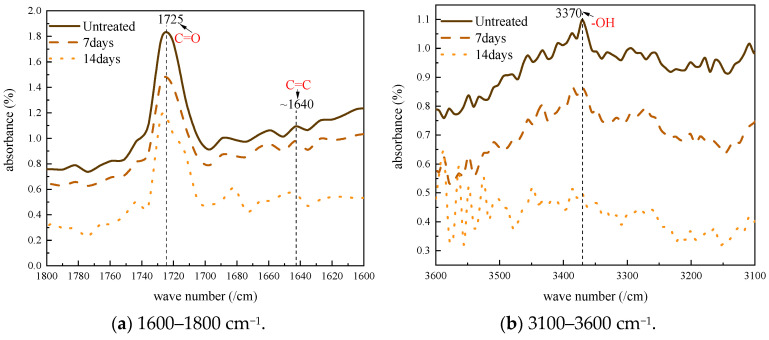
Detailed FTIR spectra.

**Figure 3 polymers-17-00431-f003:**
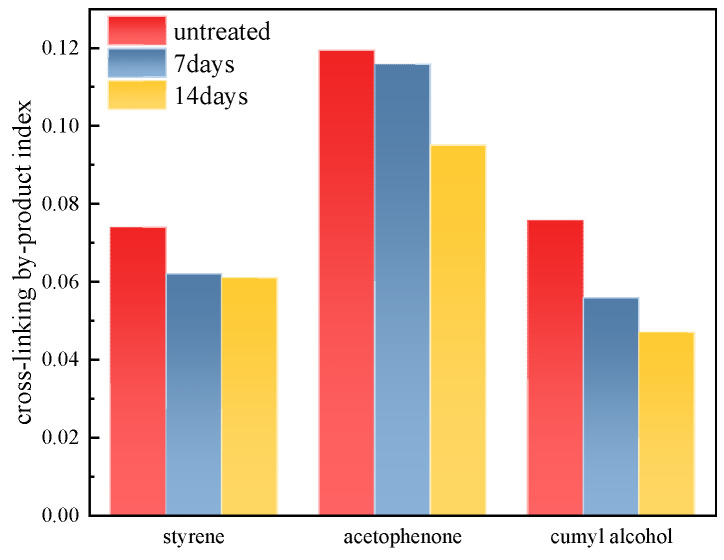
Cross-linking by-product index after treatment with different time.

**Figure 4 polymers-17-00431-f004:**
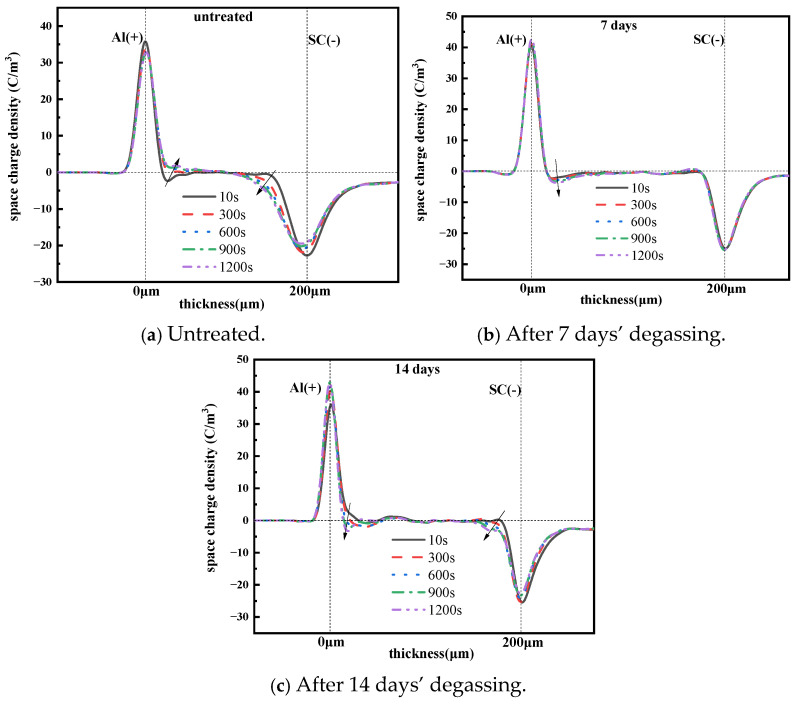
Space charge distribution of the 0.2 mm XLPE film under a 50 kV/mm electrical field.

**Figure 5 polymers-17-00431-f005:**
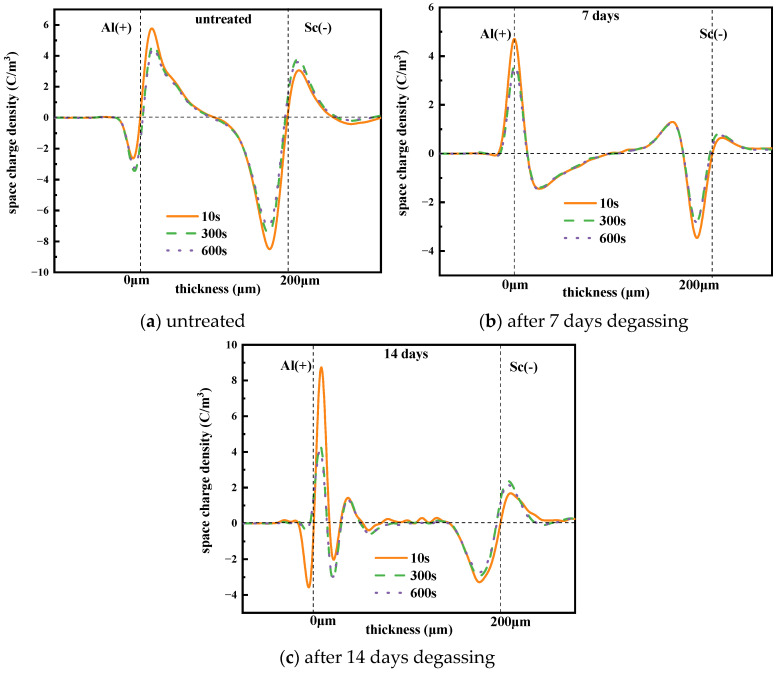
Space charge distribution during discharging process.

**Figure 6 polymers-17-00431-f006:**
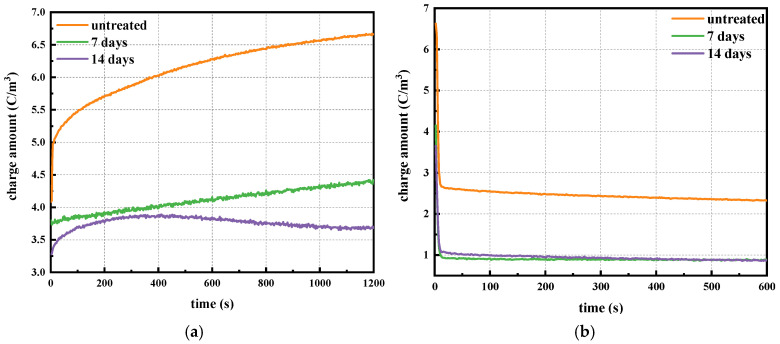
Normalized charge density during charging and discharging. (**a**) Normalized charge density during charging; (**b**) normalized charge density during discharging.

**Figure 7 polymers-17-00431-f007:**
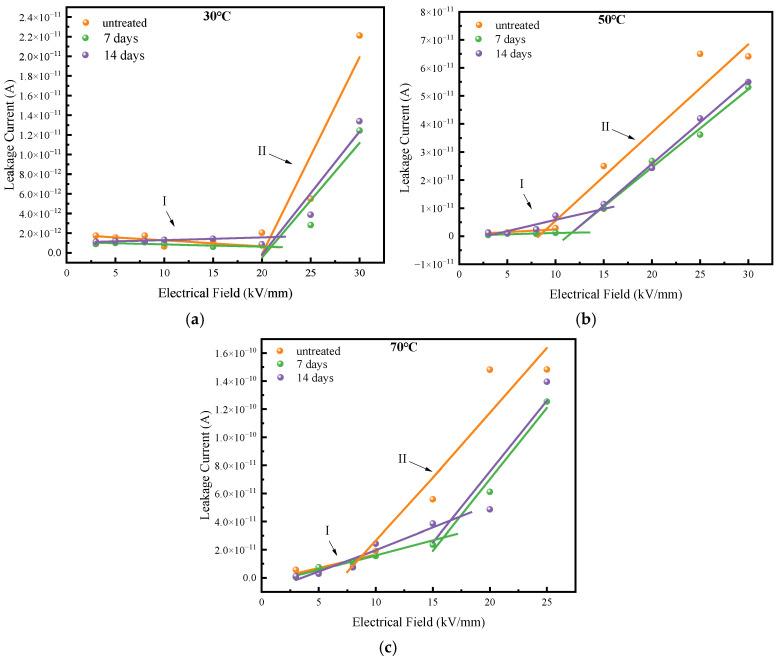
Leakage current of XLPE films. (**a**) Leakage current of XLPE film at 30 °C. (**b**) Leakage current of XLPE film at 50 °C. (**c**) Leakage current of XLPE film at 70 °C.

**Figure 8 polymers-17-00431-f008:**
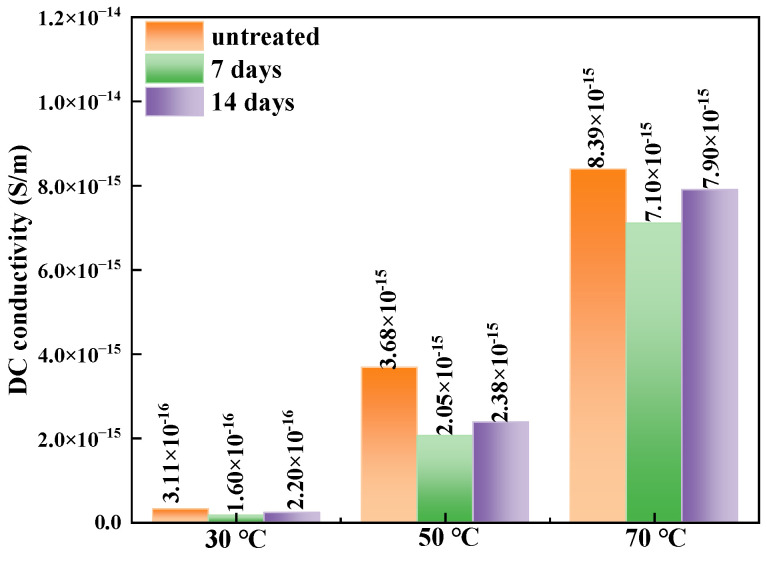
DC conductivity of each XLPE sample under 25 kV/mm electric field.

**Figure 9 polymers-17-00431-f009:**
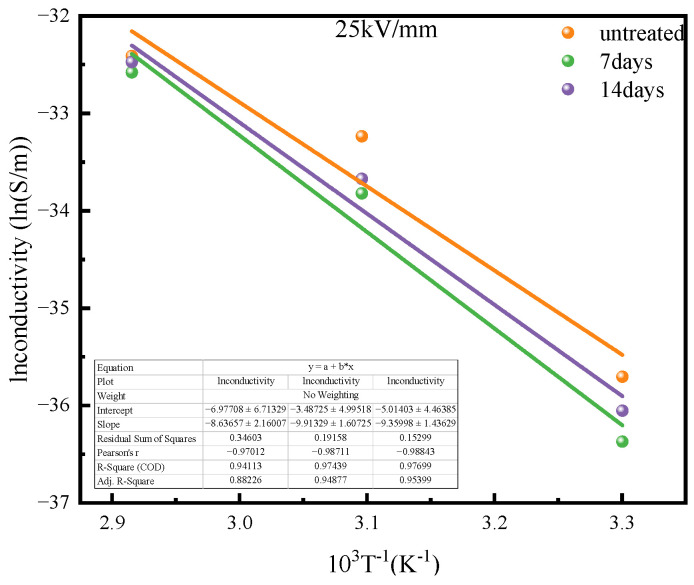
Logarithm of conductivity as a function of the reciprocal of Kelvin temperature at 25 kV/mm.

**Figure 10 polymers-17-00431-f010:**
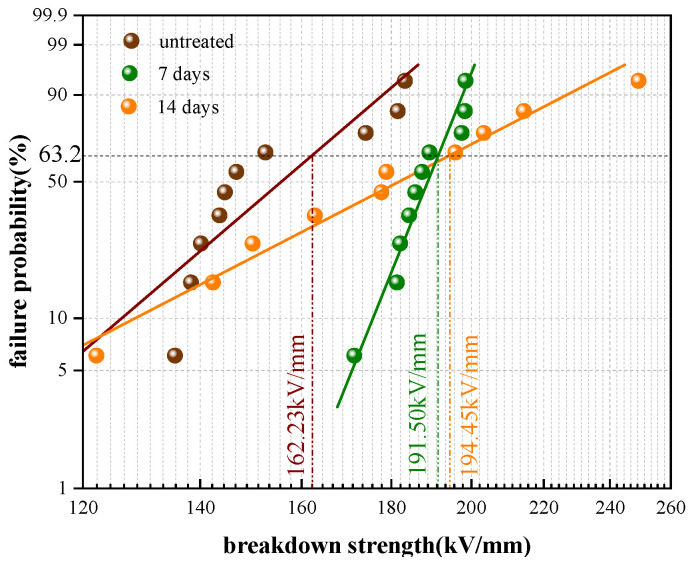
Weibull plots of DC breakdown strength of XLPE after different aging times.

**Figure 11 polymers-17-00431-f011:**
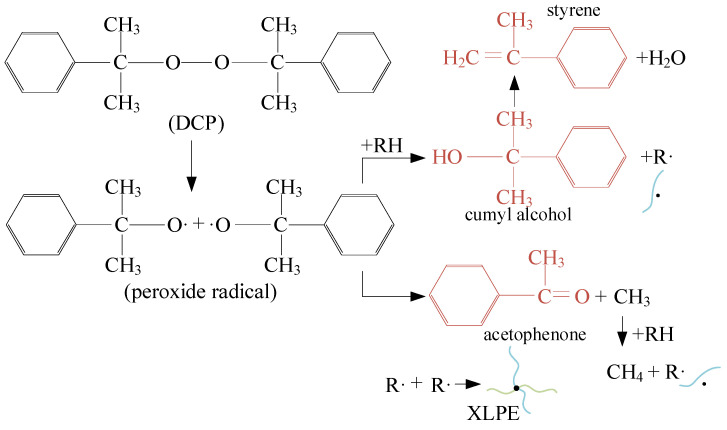
By-product generation process during crosslinking treatment.

**Table 1 polymers-17-00431-t001:** The space charge decay rate in XLPE before and after degassing treatment.

Degassing Time	Average Space Charge Decay Rate (10^−3^ C·m^−3^·s^−1^)
Untreated	7.2888
7 days	5.5299
14 days	4.7108

**Table 2 polymers-17-00431-t002:** Parameters of the fitted curves of the leakage current under each temperature.

Temperature	Samples	Slope of the Fitted Curve in Area I	Slope of the Fitted Curve in Area II	Field Threshold (kV/mm)
30 °C	untreated	−5.96 × 10^−14^	2.00 × 10^−12^	20.47
	7 days	−2.53 × 10^−14^	1.17 × 10^−12^	20.93
	14 days	2.52 × 10^−14^	1.25 × 10^−12^	21.44
50 °C	untreated	1.87 × 10^−13^	3.14 × 10^−12^	8.89
	7 days	9.28 × 10^−14^	2.79 × 10^−12^	11.67
	14 days	8.07 × 10^−13^	2.96 × 10^−12^	14.49
70 °C	untreated	1.93 × 10^−12^	9.24 × 10^−12^	8.86
	7 days	2.03 × 10^−12^	1.02 × 10^−11^	15.83
	14 days	3.02 × 10^−12^	1.01 × 10^−11^	16.35

## Data Availability

The data presented in this study are available on request from the corresponding author due to privacy.

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
