# Peer review of "Effects of Degassing Treatment on the Dielectric Properties of XLPE Insulation Used in High-Voltage DC Power Cables"

_polymers, 2025, doi:10.3390/polym17030431_

Round 1

Reviewer 1 Report

Comments and Suggestions for Authors

1. The introduction is presented comprehensively but it could be more concise for better understanding. In my view, though the background on XLPE properties and challenges is detailed clearly, some details like chemical cross-linking mechanisms may not be directly relevant to the specific scope of degassing. Further, it lacks clarity in presenting the research gap and the objectives. Explicitly stating what remains unknown about degassing effects on XLPE insulation would strengthen the motivation for the study. It is recommended to focus more on the current challenges and gaps in degassing research to better frame the study's novelty.

2. As for as the literature concerned, the manuscript reviewed several references but lacks a consistent review to highlight the novelty. It briefly touches on prior works but doesn’t adequately distinguish this study from them. Further, it is found that the most references are very old, with a few from the early 2000s. Though the foundational works are important, it is suggested to review and discuss the recent studies to present the existing state of the art to the readers.

3. The experimental procedures are clearly presented; however, the justification for selecting 7 and 14 days as degassing durations is not well-explained in the text. It can be comprehended by providing a rationale for these durations and discuss whether intermediate time points (e.g., 3 or 10 days) might offer additional insights.

4. Also, a few analyses are superficially presented. For example, the connection between by-product reduction and charge transport mechanisms could be elaborated in detail. It is recommended to strengthen the discussion of the results, particularly linking molecular-scale changes (by-products) to macro-scale properties like leakage current and dielectric strength.

5. Figures 2 and 3 seem to have overlapping content that could be combined or simplified. Figures, such as Figure 1 (FTIR spectra), are not fully explained in the text. For instance, the implications of specific absorption peaks and their relation to by-product content should be elaborated. The redundant graphs can be combined/consolidated to improve clarity and to improve the understanding of the reader.

6. Although Weibull analysis is used, other statistical methods (e.g., ANOVA) to compare untreated, 7-day, and 14-day samples could be used to validate findings more rigorously.

7. Terms like "hetero space charges" and "homo space charges" are used but not defined clearly in the text, which could confuse readers unfamiliar with these concepts.

8. The discussion part seems to simply restate results without presenting deeper insights into mechanisms or broader implications. It is recommended to provide a more theoretical explanation of the observed trends, particularly the slight increase in conductivity after 14 days.

9. There are minor grammatical errors and complex phrasing (e.g., “the charge amount trapped by sallow traps resulting from by-product”). It is suggested to proofread the manuscript for grammar and enhance the clarity of the sentences to improve readability.

10. The conclusion seems to simply restates results but doesn’t emphasize the broader significance of the findings. Highlight the important findings and include the scope of future work.

Comments on the Quality of English Language

There are minor grammatical errors and complex phrasing (e.g., “the charge amount trapped by sallow traps resulting from by-product”). It is suggested to proofread the manuscript for grammar and enhance the clarity of the sentences to improve readability.

Reviewer 2 Report

Comments and Suggestions for Authors

Paper 3 378 114 Polymers

1. The paper describes the problem of degassing the XLPE insulation of high voltage DC power cables. The authors described briefly the state-of-art connected with this subject, that can be found in technical literature. The degassing procedure of XLPE insulation in power cables is necessary to eliminate the by-products of the cross-linking process used in cables manufacturing.      

2. The paper, in literature review, contains the theoretical description of the problem and approach to this subject presented by other researchers in their papers. The authors found that the time of 7 days of degassing procedure, is enough to obtain the sufficient parameters of insulating material. The obtained results are clearly described and given in graphs, formulas, algorithms and tables. I cannot see the weak points of the paper.

3. Some minor/major recommendations/remarks should be taken into consideration for paper improvement, namely:

1.      Paper is well organized and prepared almost in accordance with mdpi magazines template.

  1. Lines 12, 20 and many others – is “90 oC” – should be “90oC” – without the space. Please correct this in the whole text, e.g. lines 87, 88, 91 etc.  
  2. Line 83 – is “2.1 sample…” – should be:  “2.1. Sample…”. Please correct this in other parts of the text, e.g. line 95 and others.
  3. Line 87 – is “15 Mpa” – should be “15 MPa”.
  4. Line 114 – is “…ball-ball…” – should be “…sphere-to-sphere..”.
  5. Line 120 – should be “Fourier”.
  6. Lines 121 and 122 – is “…718cm-1…”  – should be “…718  cm-1…” – the space/distant between the number and its unit – like it is in other lines. Please correct this in the whole paper (e.g. Table1 etc).
  7. Line 140 – Number of this figure is wrong! Should be Figure 2. Moreover should be upper case “D” in Figure 1. caption. This same in other captions below the figures. Please correct the numbers of other figures in the text!!!
  8. Line 180 – Number of this figure – should be Figure 4!!! Moreover – what is Al(+) and SC(-) in this graph? Sc(-) in Figure 5.
  9. Formula (2) between lines 227 and 228 – there is two times “t2” in this formula. Shouldn’t be “t1” in the first bracket?
  10. Line 279 – should be “DC” (upper case letters) as it is in other parts of the text. Please find this and correct in the whole text.
  11. Line 293 – put the unit of the Boltzmann coefficient [J/K] or [eV/K], because the authors inserted the units in description of other variables.
  12. Line 302 – Kelvin
  13. Line 353 inside the Figure 11 – should be “alcohol”

Reviewer 3 Report

Comments and Suggestions for Authors

The manuscript (polymers-3378114) presents the characterization of crosslinked polyethylene for use as electric cables insulation. The authors chose to investigate the influence of crosslinking by products and the effect of a thermal/degassing treatment on the conductivity of the crosslinked polyethylene samples. The manuscript is well arranged and structured, and I would recommend its publication with minor revisions. Thus, I have only the following observations prior to publication:

1.     Figures are not correctly numbered.

2.     Figure 1 – too small please enlarge. To whom is attributed the peak at 1095 cm-1, Si-O?! where from and why it is degreasing due to degassing?

3.     Please present the formula to calculate the cross-linking index by-product (Figure 3). C=C can be present in the PE back bone also due to radical interactions formation of radical species and disproportionation. Further from dicumylperoxide during decomposition alfa-methylstyrene is formed (as presented in Figure 11 – not styrene). Please correct Figure 11.

4.     Figure 9 only 3 points on the graph. Considering R(2) value of 0.88 additional points may display a volcano plot shape.

5.     Figure 10 (as numbered in the manuscript) (it is not correctly numbered in the manuscript), the Weibull plot can also be interpreted as higher failure probability at lower breakdown strength values for the 14 days degassed samples (add the R squared values). This would signify that longer degassing treatment can improve the electric strength but leads to additional failure at lower values? What would be an explanation for this observation?

Comments on the Quality of English Language

Minor typing check required. Attention to figure numbering, subscript etc.

Round 2

Reviewer 1 Report

Comments and Suggestions for Authors

The authors have responded to the comments to a very limited extent. I feel that the major concerns on the manuscript have not yet been satisfactorily addressed.

Specifically, comments 1, 2, 5, and 10 are not addressed properly. I don't see any significant improvement in this revision.

Comments 4 and 8 are addressed without strong literature evidence. The discussion part still lacks a strong analysis of the obtained results. The scientific robustness of the discussion section needs to be enhanced by presenting a clear analysis of the results with facts and evidence. Their significance in comparison to the existing literature. 

The figures are placed in a scrambled manner. They are not suitably/orderly placed. It further creates an ambiguity in reading. 

Comments on the Quality of English Language

Even after the revision, the language of the paper has scope for improvement.

Round 3

Reviewer 1 Report

Comments and Suggestions for Authors

In my opinion, the authors lack in comprehending the comments appropriately and trying to convince with some limited modifications rather than improving the quality of the manuscript.

In my earlier comments I have expected the authors to critically discuss the current challenges and gaps in degassing research and present the objectives of the study as well as the novelty of the work explicitly. Further I suggested to review and discuss the recent studies to present the existing state of the art, which are published in very recent past to show the validity of the present study. It was recommended to strengthen the discussion of the results through critical analysis and with a strong comparison with the existing literature. In figure 1, it is expected to discuss the implications of specific absorption peaks and their relation to by-product. And I expected to emphasize the broader significance of the findings in the conclusion instead of simply restating them again.

However, the authors still fail to show an acceptable revision after multiple chances. I recommend the authors to put an appropriate effort for improving the manuscript as per the suggestions.

Comments on the Quality of English Language

Need improvement.
